# Major Depressive Disorder and Pulmonary Tuberculosis Comorbidity Exacerbates Proinflammatory Immune Response—A Preliminary Study

**DOI:** 10.3390/pathogens12030361

**Published:** 2023-02-21

**Authors:** Magaly Alvarez-Sekely, Ana Lopez-Bago, Renata Báez-Saldaña, Rodolfo E. Pezoa-Jares, Patricia Gorocica, Edgar Zenteno, Ricardo Lascurain, Alfredo Saldívar-González

**Affiliations:** 1Banco de Sangre, Hospital San Angel Inn Universidad, Mexico City 03330, Mexico; 2Departamento de Bioquímica, Facultad de Medicina, Universidad Nacional Autónoma de México, Mexico City 04510, Mexico; 3Clínica de Neumología Oncológica, Instituto Nacional de Enfermedades Respiratorias “Ismael Cosío Villegas”, Secretaría de Salud, Mexico City 14080, Mexico; 4Escuela de Medicina y Ciencias de la Salud, Tecnológico de Monterrey, Monterrey 64849, Mexico; 5Departamento de Investigación en Bioquímica, Instituto Nacional de Enfermedades Respiratorias “Ismael Cosío Villegas”, Secretaría de Salud, Mexico City 14080, Mexico; 6Departamento de Farmacología, Facultad de Medicina, Universidad Nacional Autónoma de México, Mexico City 04510, Mexico

**Keywords:** tuberculosis, depressive disorder, pro-inflammatory, cytokines, patients

## Abstract

Background: Major depressive disorders (MDDs) occurs frequently in patients with tuberculosis (TB). Elevated serum pro-inflammatory cytokine levels in MDD patients is a well-established fact. Therefore, an integrated clinical practice should be considered. However, the inflammatory status of MDD-TB patients is unknown. In this study, we analyze cytokines in activated-cells and sera from MDD-TB, TB, MDD patients, and healthy controls. Methods: Flow cytometry was used to evaluate the intracellular production of interferon (IFN)-gamma, tumor necrosis factor (TNF)-alpha, interleukin (IL)-12, and IL-10 by peripheral blood mononuclear cells after a polyclonal stimulation. A Bio-Plex Luminex system was used to measure serum cytokine and chemokine levels in the study groups. Results: We observed a 40.6% prevalence of MDD in TB patients. The proportion of IFN-gamma-producing cells was higher in MDD-TB patients than other pathological groups. Nevertheless, the percentage of TNF-alpha- and IL-12-producing cells was similar between MDD-TB and TB patients. Likewise, MDD-TB and TB patients showed similar serum pro-inflammatory cytokine and chemokine levels, which were significantly lower than those in MDD patients. By multiple correspondence analyses, we observed that low levels of serum IL-4, IL-10, and IL-13 were powerfully associated with TB comorbidities with MDD. Conclusions: A high frequency of IFN-γ-producing cells is associated with low levels of serum anti-inflammatory cytokines in MDD-TB patients.

## 1. Introduction

The World Health Organization (WHO) has estimated that 322 million of the world’s population is living with major depressive disorder (MDD). The MDD worldwide rate increased by 18.4% from 2005 to 2015 [1], which has attracted attention to increase the research on this disorder. A significant association between high serum inflammatory biomarker levels and MDD has been highlighted by several authors [2,3,4]. MDD is frequently comorbid with chronic and systemic medical diseases [5]. Numerous studies have also reported comorbid depression in patients with tuberculosis (TB), with prevalence ranging from 9.9 to 46.3% [6,7,8,9,10,11]. 

TB is an infectious disease mainly caused by *Mycobacterium tuberculosis* (*Mtb*), which causes high mortality rates in humans worldwide; an estimated 10.0 million new TB cases and 1.2 million TB deaths were reported in 2018 [12]. The control of *Mtb* infection requires a cell-mediated immunity induced by interleukin (IL)-12 [13]. In cell-mediated immunity, the pro-inflammatory cytokines interferon gamma (IFN-γ) and tumor necrosis factor-alpha (TNF-α) are essential to restrict bacilli within granulomas and to induce *Mtb*-infected macrophage death [14,15]. An important mechanism of the immune system’s response against *Mtb* is the downregulation exerted by IL-10, an anti-inflammatory cytokine [16]. IL-10 presents a high serum level in active TB, remaining high even at the end of anti-TB treatment [17].

Despite the relationship between inflammatory status and MDD, inflammation-related cytokine levels in depressed TB patients are unknown. The examination of pro-inflammatory molecules in patients with both TB and MDD is an essential practice for a better understanding of the pathophysiology mechanisms involved in the mediation of this comorbidity.

The aim of this study is to analyze the pro-inflammatory cytokines in activated-cells and sera from MDD-TB patients. We observed a high frequency of IFN-γ-producing cells from MDD-TB patients, which is consistent with a powerful association between low levels of anti-inflammatory cytokines and TB and MDD comorbidities.

## 2. Materials and Methods

### 2.1. Participants

The study population consisted of 32 pulmonary TB patients, who were recruited from the Instituto Nacional de Enfermedades Respiratorias “Ismael Cosío Villegas” in Mexico City. The diagnosis of TB was based on chest X-rays, clinical results, and positive Ziehl–Neelsen test results in sputum. The TB diagnosis was confirmed by *M. tuberculosis* growth in sputum culture. TB patients were classified as a class 3 category I disease, according to the American Thoracic Society [18]. MDD was diagnosed by a standard psychiatric interview, conducted by an expert, based on the criteria established by the American Psychiatry Association [19] and by both the Beck Depression Inventory (BDI) and Hamilton Depression Rating Scale (HDRS). BDI is a self-assessment tool that rates MDD symptoms by a score ≥ 17, whereas HDRS administered by an expert establishes a score ≥ 18 as positive for MDD [20,21]. A group formed of 11 patients suffering from MDD with no TB was recruited from the Instituto Nacional de Psiquiatría “Ramón de la Fuente Muñiz” in Mexico City, diagnosed in a similar way as the other groups described above. The specific enrolment criteria were defined as adult individuals and the absence of other concomitant diseases, such as autoimmune, endocrine, and metabolic diseases, atopy, immunodeficiency, and cancer. The control group included 11 unrelated, healthy volunteers who had no history of chronic illnesses or atopy. All control individuals were tuberculin-reactive with clinical evidence of a bacillus Calmette–Guérin (BCG) scar, since they had received the BCG vaccine during childhood. Additionally, the World Health Organization Quality of Life (WHOQOL) survey was administered to each participant [22]. General data from the study groups are shown in Table 1. All participants signed their informed consent forms for inclusion, before they participated in the study. The study was performed according to the Declaration of Helsinki, and the protocol was approved by the Medical Ethics Committee of the Instituto Nacional de Enfermedades Respiratorias “Ismael Cosío Villegas” by way of identification code B34-10. 

### 2.2. Antibodies and Reagents

Fluorescein isothiocyanate (FITC)-labeled mouse monoclonal antibodies (mAbs) for human CD69, CD25, IFN-γ, and TNF-α; phycoerythrin (PE)-labeled mAbs for CD25, CD69, IL-12, and IL-10; and FITC- and PE-labeled isotype control mAbs were purchased from BD Biosciences Pharmingen (San Diego, CA, USA). RPMI-1640 culture medium, trypan blue dye, bovine serum albumin fraction V (BSA), phorbol 12-myristate 13-acetate (PMA), brefeldin-A, saponin, *p*-formaldehyde, and salt reagents were obtained from the Sigma-Aldrich Company (St. Louis, MO, USA). Bio-Plex Pro^TM^ human cytokine, chemokine, and growth factor kit for 16-plex was acquired from Bio-Rad Laboratories, Inc. (Hercules, CA, USA).

### 2.3. Blood Samples

Heparinized whole blood (10 mL) obtained from each participant was centrifuged at 350× *g* for 10 min, and serum was frozen at −30 °C until used. A blood cell pellet was suspended in phosphate-buffered saline (PBS; 10 mM sodium phosphate, 150 mM sodium chloride, pH 7.2) up to the original volume. Peripheral blood mononuclear cells (PBMCs) were isolated from PBS-suspended blood cells by Lymphoprep (Axis-Shield PoC As, Oslo, Norway) 1.077 density gradient centrifugation for 30 min at 500× *g*, 16 °C [23]. Following centrifugation, the interface cells were collected, washed in PBS, and counted in a Neubauer chamber, assessing the cellular viability by the trypan blue-dye exclusion method.

### 2.4. Cell Culture and Flow Cytometry

PBMCs (2 × 10^5^ cells/mL) were stimulated by PMA (50 ng/mL) plus ionomycin (1 μg/mL) for 24 h in 96-well flat-bottomed cell culture plates (Corning Costar Sigma-Aldrich) in RPMI-1640 supplemented with 1 mM of sodium pyruvate, 2 mM of *L*-glutamine, 50 μM of 2-mercaptoethanol (Gibco BRL, Rockville, MD, USA), 100 IU/mL of penicillin, 100 μg/mL of streptomycin, and 10% heat-inactivated fetal calf serum (Hyclone Laboratories Inc., Logan, UT, USA) at 37 °C in a 5% CO_2_ humidified atmosphere. Brefeldin-A was added (10 μg/mL) 6 h before the end of the stimulation period. Cells were harvested, washed in PBS containing 0.2% BSA and 0.2% sodium azide (PBS-BSA buffer), and stained by either FITC-labeled anti-CD69 mAb or PE-labeled anti-CD25 mAb in the dark for 30 min at 4 °C. Then, the cells were washed twice in PBS-BSA buffer, fixed in 4% *p*-formaldehyde in PBS for 10 min at 4 °C. Cells were washed twice in PBS and permeabilized in saponin buffer (0.1% saponin, 0.01% pig IgG, 10 mM HEPES, 10% BSA in PBS), and gently shaken for 10 min. Subsequently, the cells were stained by fluorescent mAbs to TNF-α or IFN-γ, IL-10, and IL-12, depending on the antibody combinations. Following 30 min of incubation, the cells were washed with saponin-washing buffer (0.1% saponin, 100 mM HEPES in PBS), suspended in a FACS flow buffer, and analyzed by flow cytometry. In each case, 10,000 cells were acquired in a FACSCalibur flow cytometer (BD Biosciences, San Jose, CA, USA). Cells were counted in linear mode for side scatter and log amplification for cell-activation marker expressions CD69 and CD25 on PBMC. Preliminary analyses showed that the percentage of CD25^+^ cells was lower than those of CD69^+^ cells after being activated by PMA plus ionomycin. Therefore, CD69^+^ cells were then gated and analyzed for intracellular cytokine expression, as described above. Fluorochrome-labeled isotype-matched control mAbs were used to evaluate background staining.

### 2.5. Cytokine Measurement

The serum cytokine levels were determined by a 16-plex magnetic bead-based antibody detection kit, according to the manufacturer’s instructions (Bio-Rad Labs, Hercules, CA, USA). The plate was read on a Bio-Plex Luminex 200 system and the cytokine concentrations using standards were calculated in pg/mL by Bio-Plex Manager 6.0 software (Bio-Rad Labs).

### 2.6. Statistical Analysis

The data were evaluated by D’Agostino and Pearson’s omnibus test for normal distributions [24]; whereas the homogeneity of variance was assessed by Bartlett´s test [25]. For non-parametric distributed variables, a Log_10_ transformation was performed to obtain data normality. For symmetrical distributions, a one-way ANOVA followed by the Dunnett´s multiple comparison test were used to evaluate the differences between the mean values of each pathological and control group [26,27]. Moreover, one-way ANOVA followed by Tukey´s D’Agostino honest significance difference test was used to compare all the possible pairs of pathological conditions [28]. When appropriate, the non-parametric Kruskal–Wallis ANOVA, followed by Dunn’s multiple comparison test, was used. The values are shown as the mean ± standard error of the mean (SEM) and differences among groups were considered statistically significant at *p* ≤ 0.05. All the statistical analyses and graphs were composed in GraphPad Prism^TM^ 7.0 (GraphPad Inc., La Jolla, CA, USA). Additionally, a multiple correspondence analysis [29] was used to evaluate the association pattern of groups (TB, MDD, MDD-TB, and control) and the levels (low, medium, and high) of each type of serum cytokine (pro-inflammatory, anti-inflammatory, and chemokines). The results from this analysis are shown as a simplified graphical output, which displays a cluster of points positioned between two dimensions (Dim 1 and 2). The multivariate analyses were performed using the Proc Corresp statement of the Statistical Analysis System (SAS) University software (SAS Institute, Cary, NC, USA). The analysis of the Beck depression inventory, Hamilton depression rating scale, and World Health Organization Quality of Life were achieved by one-factor ANOVA, followed by the Bonferroni post hoc test. 

## 3. Results

### 3.1. Mayor Depressive Disorder Was Frequent in TB Patients

We observed that 40.6% (13/32) of TB patients met the criteria for MDD. In Table 1 shows that there was no significant difference among all the groups regarding age; the mean age was 40.4 ± 15.1 years. TB and MDD-TB patients and healthy controls showed similar distributions of gender: 48.8% men and 51.2% women. The WHOQOL survey exhibited that controls and TB patients obtained high scores. Nonetheless, the depressive condition reduced by a 1.5-fold score in quality of life as compared with other pathologies. In Table 1, the MDD-TB group shows a 1.49- and 1.30-fold reduction as compared with the control and TB groups, respectively, whereas both BDI and HDRS scales obtained positive scores for MDD.

**Table 1 pathogens-12-00361-t001:** General data from patients and healthy controls.

	Healthy Controls (*n* = 11)	MDD Patients (*n* = 11)	TB Patients (*n* = 19)	MDD-TB Patients (*n* = 13)
Age (years)	28.9 ± 3.6	35.5 ± 9.0	44.2 ± 15.0	49.5 ± 18.8
Gender				
Men	5 (45.5%)	0 (0.0%)	8 (42.1%)	8 (66.7%)
Women	6 (54.5%)	11 (100%)	11 (57.9%)	5 (33.3%)
BDI score	1.9 ± 3.1	38.4 ± 11.5	4.5 ± 2.0	20.8 ± 3.3
HDRS score	1.0 ± 1.7	24.4 ± 4.8	3.4 ± 2.0	24.8 ± 3.0
WHOQOL score	98.8 ± 5.4	65.6 ± 16.4	92.7 ± 12.8	66.3 ± 14.5

TB, tuberculosis; MDD, major depressive disorder; BDI, Beck depression inventory; HDRS, Hamilton depression rating scale; WHOQOL, World Health Organization Quality of Life. The results are presented as mean ± SD values.

### 3.2. High Intracellular IFN-γ-Producing Activated Cell Percentage in MDD-TB Patients

The percentage of IFN-γ-producing cells in MDD-TB patients was the highest among the pathological groups (Figure 1A). By multiple correspondence analysis, a high quantity of IL-12- and IFN-γ-positive cells exhibited a strong association with the MDD-TB patients (Figure 2). Concerning the TNF-α- and IL-12-positive cells, both MDD-TB and TB groups showed comparable proportions (Figure 1B,C). In contrast, the proportion of IL-12-positive cells from MDD patients was the lowest among all groups (Figure 1C, *p* < 0.05). Moreover, the low percentages of IL-10-, IL-12-, TNF-α-, and IFN-γ-positive cells were strongly associated with MDD (Figure 2).

### 3.3. Low Levels of Serum Anti-Inflammatory Cytokines Were Strongly Associated to the MDD-TB Group

Regarding the serum pro-inflammatory cytokine quantity (pg/mL), *Mtb*-infected patients showed similar IFN-γ, IL-17, TNF-α, IL-1β, IL-6, and IL-12 levels (Figure 3). However, the serum TNF-α level in TB patients was lower than in the controls (Figure 3C, *p* < 0.05). Additionally, by the multiple correspondence analyses, no association of pro-inflammatory cytokines with *Mtb*-infected patients was observed (Figure 4A).

Concerning the anti-inflammatory cytokines, *Mtb*-infected patients also displayed comparable levels of serum IL-4, IL-10, and IL-13, but the final cytokine value was lower in MDD-TB patients than in the controls (Figure 5C, *p* < 0.05). Interestingly, a strong association was observed between a low concentration of anti-inflammatory cytokines and the cluster formed by MDD-TB patients (Figure 4B).

With respect to MDD patients, the serum levels of IFN-γ, IL-17, and IL-10 were the highest among the analyzed groups (Figure 3A,B and Figure 5B, *p* < 0.05), but the TNF-α level was the lowest in these groups (Figure 3C, *p* < 0.05). Likewise, a high amount of IFN-γ, IL-17, and IL-1β levels in the serum presented a strong association with MDD patients (Figure 4A). Similarly, high levels of IL-10 and IL-13 also showed an association with MDD patients. In addition, medium levels of IL-4, IL-10, and IL-13 had an association to TB (Figure 4B).

### 3.4. Serums MCP-1β and IL-8 Showed Similar Levels in Mtb-Infected Patients

An analysis of pro-inflammatory chemokines and IL-7 showed that their levels were always similar in TB and MDD-TB patients. The serum levels of MCP-1(CCL2), MIP-1β (CCL4), and IL-8 (CXCL8) were lower in both TB and MDD-TB groups than in MDD patients (Figure 6A–C, *p* < 0.05). In contrast, sera from MDD patients exhibited the highest amount of MIP-1β and the lowest level of IL-7 than sera from the other analyzed groups (Figure 6, *p* < 0.05).

## 4. Discussion

Numerous studies have reported a frequent association between MDD and TB [6,7,8,9,10,11]. We observed an important proportion of TB patients who met the criteria for MDD (40.6%), which was similar to 46.3% reported in Pakistan, 45.5% in Nigeria, and 43.4% in Ethiopia, using different scales [6,8,11]. Conversely, Moussas et al. observed a 9.9% depression rate in TB patients from Greece [7], whereas Matsumoto et al. reported a depressive state in 16.8% of TB patients [10]. These differences, depending on the social characteristics of the studied population, applied scales in each of the methodological designs, such as poor social status, deficient quality of life, and deprivation of health service supports. 

Because the inflammatory status of MDD-TB patients is unknown, we first analyzed the frequency of peripheral blood cytokine-producing cells following an in vitro polyclonal stimulus. Then, we analyzed the serum levels of cytokines and chemokines in the study groups. A similar frequency of cytokine-producing cells between TB and MDD-TB patients was observed, which was higher than that for MDD patients. The explanation could be due to a large proportion of antigen-committed cells that circulate through secondary lymphoid organs and damaged pulmonary tissue from *Mtb*-infected patients [30]. However, TB and MDD-TB patients showed low levels of serum pro-inflammatory molecules as compared with MDD patients, which may be because mycobacterial infection induces an inflammatory response at the site of lung damage, whereas in MDD, it seems to be a systemic inflammatory condition.

Likewise, the serum levels of IFN-γ, IL-17, IL-10, MCP-1, IL-8, and MIP-1β were similar in TB and MDD-TB patients, but significantly lower as compared with MDD patients. Our results show that serum levels of IFN-γ, IL-17, and MIP-1β were the highest in depressive patients than in *Mtb*-infected patients or healthy controls. Possibly, MDD patients have a systemic inflammatory status, as previously suggested [2]. MDD patients also showed the highest level of serum IL-10, which is consistent with controlling inflammatory responses. Furthermore, depressive patients suffer from oxidative stress, which is involved in inflammatory processes affecting the entire organism; it strongly supports the idea that MDD should be considered not merely as a mental disorder, but truly as a systemic disease [31].

A seemingly inconsistent fact was that MDD patients showed high serum levels of pro-inflammatory molecules, but a low proportion of pro-inflammatory cytokine-producing cells. This suggests that there are other cells that contribute to producing pro-inflammatory molecules.

Concerning TB and MDD-TB patients, the serum level of TNF-α was higher than that in MDD patients. TNF-α is an important molecule for immune responses against *Mtb*; this molecule is required for the formation of granulomas that contain bacilli, avoiding its spread [15]. Similarly, serum IL-7 was increased in TB and MDD-TB patients; this cytokine has been shown to play an essential role in the formation of germinal centers for immune responses against *Mtb* [32]. The increase in IL-7 in *Mtb*-infected patients would help to control mycobacterial growth. On the contrary, in MDD patients, the serum level of IL-7 was low, which is consistent with a previous report [33]. Maybe MDD patients do not require IL-7 because, apparently, they do not have an infectious process.

An interesting result was that MDD-TB patients showed the highest percentage of IFN-γ-producing cells. Likewise, a high proportion of IL-12- and IFN-γ-producing cells were strongly associated with the MDD-TB cluster. Furthermore, we observed in the sera that low levels of IL-10, IL-4, and IL-13 were associated with clusters of TB with MDD. This unexpected result suggests that the depressive condition modifies the production of anti-inflammatory cytokines in TB patients, which could have immunopathological consequences. The increased frequency of IFN-γ-producing cells from MDD-TB patients might be indirect due to a deficient regulation caused by low levels of anti-inflammatory cytokines, such IL-10, IL-4, and IL-13. 

Considering the evidence of MDD in patients with chronic infectious diseases, mental health should be assessed in routine clinical care, due to the close relationship between mental processes and immune response. Finally, our study presented several limitations. Cytokines secreted in the PBMC culture supernatant after activating either the polyclonal stimulus or mycobacterial antigen are unknown. The absolute numbers of cytokine-producing cells are also unknown. Further studies should be conducted to achieve a better understanding of the comorbidity of MDD with TB.

## 5. Conclusions

A high frequency of IFN-γ-producing stimulated cells from MDD-TB patients was associated with low serum levels of anti-inflammatory cytokines from MDD-TB patients. The prevalence of MDD and TB represents a frequent comorbid condition. The clinical recognition of MDD in TB patients would be suitable for the diagnosis, treatment, and prognosis of these two pathological conditions. 

## Figures and Tables

**Figure 1 pathogens-12-00361-f001:**
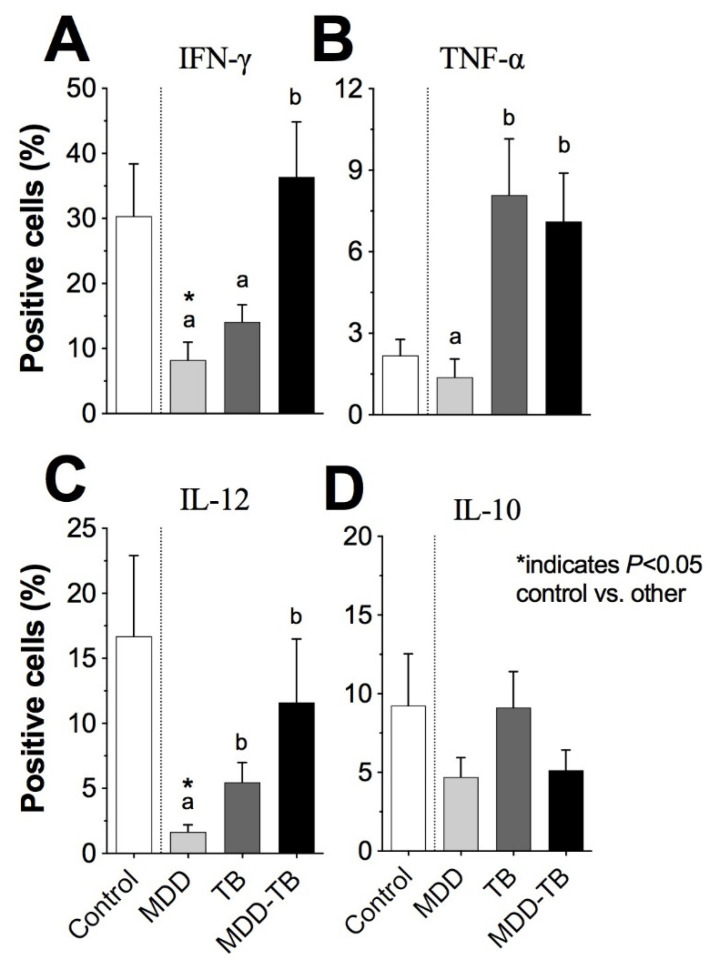
Proportion of cytokine-producing cells from patients and healthy controls. Peripheral blood mononuclear cells were stimulated by phorbol myristate acetate plus ionomycin for 24 h at 37 °C. Brefeldin-A was added 6 h before the end of the culture. Then, cells were fixed, permeabilized, stained by fluorescent mAbs to CD25 or CD69 and cytokines, and analyzed by flow cytometry. The bars denote the percentage of positive cells to IFN-γ (**A**), TNF-α (**B**), IL-12 (**C**), and IL-10 (**D**) in healthy individuals (control) and patients with major depressive disorder (MDD), tuberculosis (TB), and major depressive disorder in comorbidity with tuberculosis (MDD-TB). A star plot represents the proportions of positive cells to analyzed cytokines in study groups. The literals (a and b) indicate statistic differences at *p* < 0.05 among pathological groups. The sign (*) points out statistic differences at *p* < 0.05 between control and indicated pathological groups.

**Figure 2 pathogens-12-00361-f002:**
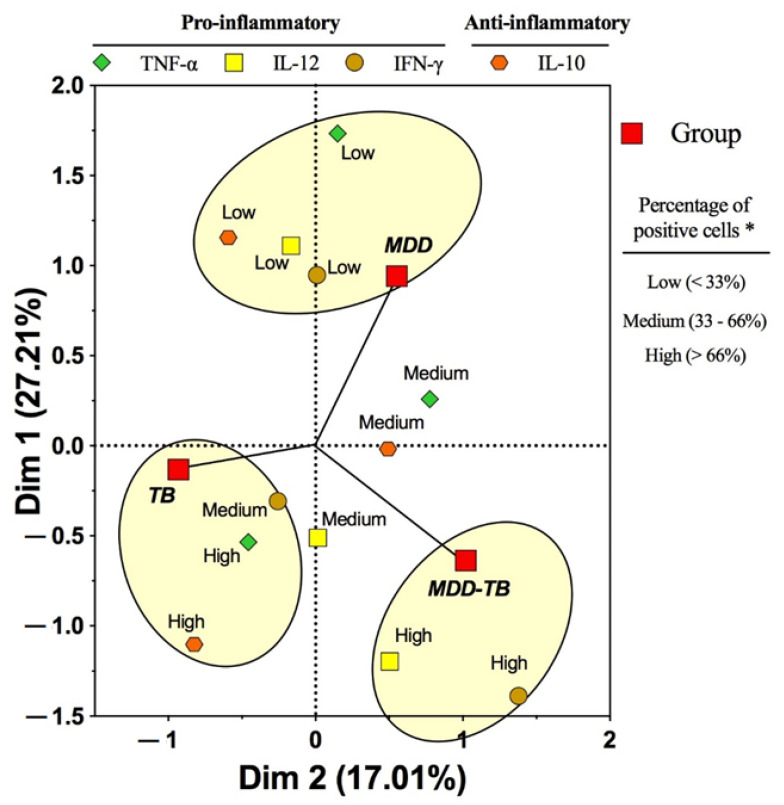
Pattern of association among pathological groups and positive cells to pro- and anti-inflammatory cytokines by means of a multiple correspondence analysis method. Peripheral blood mononuclear cells were stimulated by phorbol myristate acetate plus ionomycin for 24 h at 37 °C. Brefeldin-A was added 6 h before the end of the culture. Then, the cells were fixed, permeabilized, stained by fluorescent mAbs to CD69 and cytokines, and analyzed by flow cytometry. * The category of the percentage of positive cells was based on the 33% and 66% percentile distributions of the cytokine-positive cells evaluated by flow cytometry in the healthy control group. Arbitrary ellipses were drawn to assist the interpretation of the clusters of categorized variables. The strength of the association between variables can be interpreted in relation to the distance separating two adjacent points. Thus, at a shorter distance, the association is stronger. Positive and negative centroid coordinates for dimensions 1 (Dim 1) and 2 (Dim 2) arise from the combination of the associated categories of the variables.

**Figure 3 pathogens-12-00361-f003:**
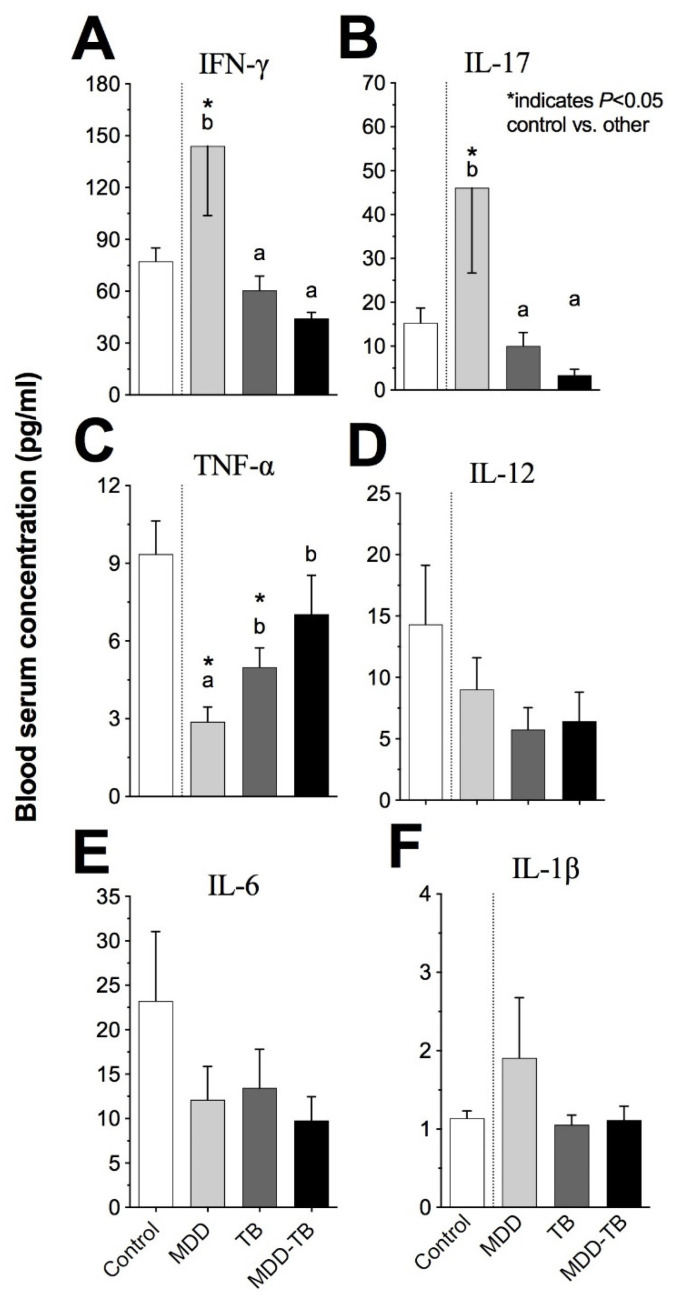
Blood serum pro-inflammatory cytokine levels. Blood serum was analyzed by a Bio-Plex Luminex system. The bars denote the serum concentrations of IFN-γ (**A**), IL-17 (**B**), TNF-α (**C**), IL-1β (**D**), IL-6 (**E**), and IL-12 (**F**) in healthy individuals (control) and patients with major depressive disorder (MDD), tuberculosis (TB), and major depressive disorder in comorbidity with tuberculosis (MDD-TB). The literals (a and b) indicate statistic differences at *p* < 0.05 among pathological groups. The sign (*) points out statistic differences at *p* < 0.05 between control and indicated pathological groups.

**Figure 4 pathogens-12-00361-f004:**
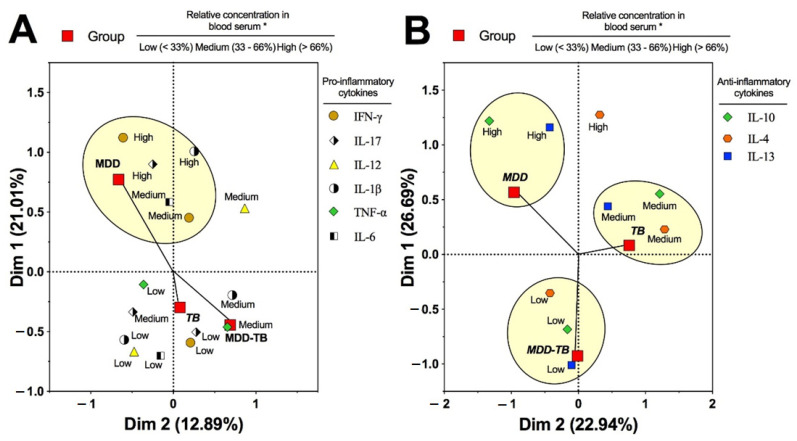
Pattern of association among pathological groups and serum concentration of pro- and anti-inflammatory cytokines by the multiple correspondence analysis method. * The category of the quantity of serum cytokine was based on the 33% and 66% percentile distributions of the evaluated cytokines by the Bio-Plex Luminex system in the healthy control group. Arbitrary ellipses were drawn to assist the interpretation of clusters of associated categories. The strength of the association between variables can be interpreted in relation to the distance separating two adjacent points. Thus, at a shorter distance, the association is stronger. Positive and negative centroid coordinates for dimensions 1 (Dim 1) and 2 (Dim 2) arise from the combination of the associated categories of the variables. High serum levels of pro-inflammatory cytokines were clustered with major depressive disorder (MDD), while that tuberculosis (TB) and MDD-TB comorbidity failed to form a cluster (**A**). High and medium serum levels of anti-inflammatory cytokines were clustered with MDD and TB, respectively. In contrast, low serum levels of these cytokines were clustered with MDD-TB (**B**).

**Figure 5 pathogens-12-00361-f005:**
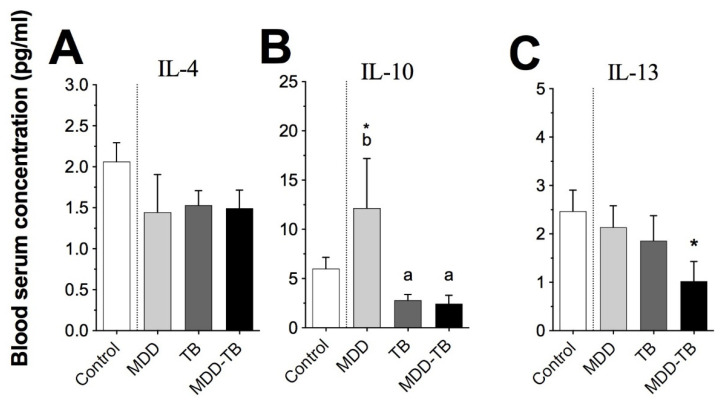
Blood serum anti-inflammatory cytokine levels. Blood serum was analyzed by the Bio-Plex Luminex system. The bars denote the serum concentrations of IL-4 (**A**), IL-10 (**B**), and IL-13 (**C**) in healthy individuals (control) and patients with major depressive disorder (MDD), tuberculosis (TB), and major depressive disorder in comorbidity with tuberculosis (MDD-TB). The literals (a and b) indicate statistic differences at *p* < 0.05 among pathological groups. The sign (*) points out statistic differences at *p* < 0.05 between the control and indicated pathological groups.

**Figure 6 pathogens-12-00361-f006:**
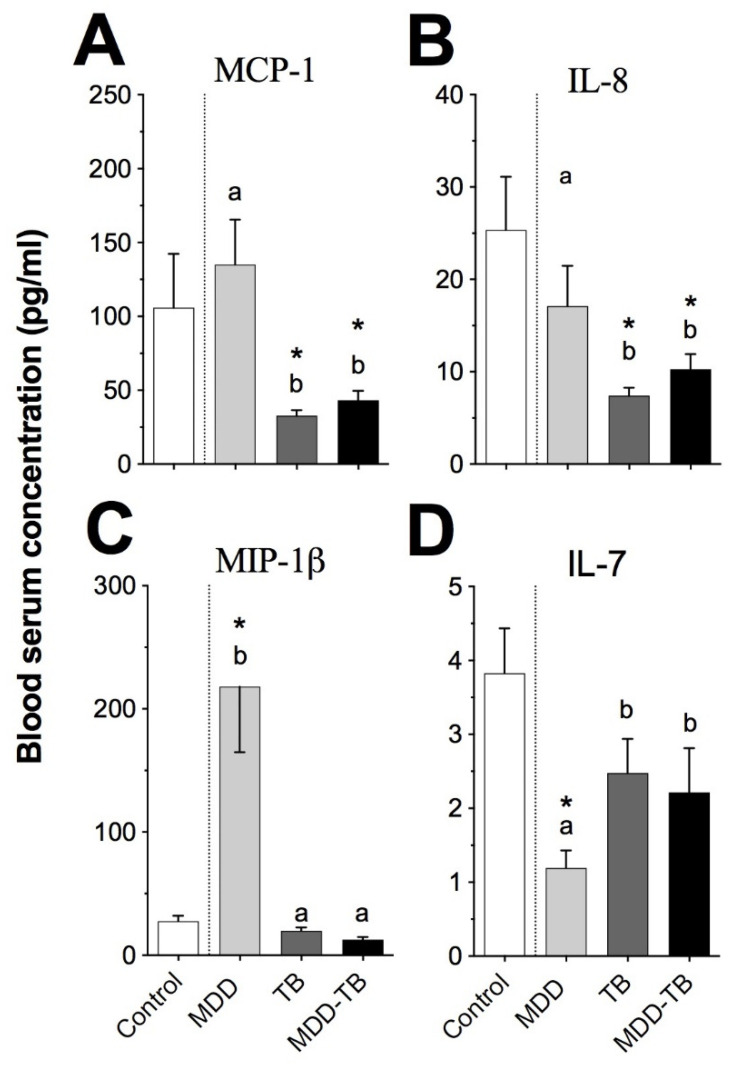
Measurements of chemokines in blood serum analyzed by the Bio-Plex Luminex system. The bars indicate the serum concentrations of MCP-1 (**A**), IL-8 (**B**), MIP-1β (**C**), and IL-7 (**D**) in healthy individuals (control) and patients with major depressive disorder (MDD), tuberculosis (TB), and major depressive disorder in comorbidity with tuberculosis (MDD-TB). The literals (a and b) point out the statistic differences at *p* < 0.05 among pathological groups, whereas the sign (*) denotes the statistic differences at *p* < 0.05 between control and indicated pathological groups.

## Data Availability

Data is unavailable due to the Mexican ethical restriction NOM-004-SSA3-2012.

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
