# Peer review of "Major Depressive Disorder and Pulmonary Tuberculosis Comorbidity Exacerbates Proinflammatory Immune Response—A Preliminary Study"

_pathogens, 2023, doi:10.3390/pathogens12030361_

Round 1

Reviewer 1 Report

The manuscript Alvarez-Sekely M. and coworkers addresses the cytokine-producing mononuclear cells and cytokine levels from blood of patients with major depressive disorders (MDD) and tuberculosis (TB) comparing them with MMD patient, TB patient and healthy subject samples.

I. Abstract: it is suitable. However, I suggest that conclusion be rewritten because I understood that the last paragraph from Introduction described better those findings of the present study. In addition, it would be interesting to comment briefly in the Abstract the importance of the study in the research field.

II. Introduction and Discussion: good and objective.

III. Material and Methods:

1. How the number of patients and healthy controls were defined for the casuistic?

2. Were PBMC stained with live dead to exclude viable from non-viable cells assessed by flow cytometry?

IV. Results:

3. The authors could show the gate strategy used for flow cytometry, and a representative analysis obtained from the more significant results.

4. Why did the authors use CD69 and CD25 staining? They did not show that results and did not mention about them. Please, explain.

5. Bar graphs should be represented also by individual data plus bar.

6. Although PBMC from MMD-TB exhibited higher IFN-gamma-producing cells compared to MMD or TB samples, those frequencies were not different from PBMC from healthy donors. How do the authors explain that? In the Discussion, the authors described: “The increased frequency of IFN-gamma-producing cells from MDD-TB patients might be indirectly due to a deficient regulation caused by low levels of anti-inflammatory cytokines, such IL-10, IL-4, and IL-13. An excess of inflammation can contribute to increased tissue damage during the immune response against mycobacterial infection [34].” Might the authors describe “an excess of inflammation” when the cytokine levels in the serum from MMD-TB patients were lower than those in MMD and similar to TB patients?

7. How do the authors explain high cytokine-producing cells and low level of serum cytokines in the MMD-TB group?

8. The results are difficult to interpret because assays were performed with polyclonal stimulation and not with mycobacteria-specific antigen. This point should be discussed. Authors need to discuss the limitations of the study.

V. Conclusion: too extensive.

9. The authors could address straight the take home message.

10. The findings do not show that: “However, the exception was a high frequency of IFN-gamma-producing stimulated cells from MDD-TB patients, which was consistent with a powerful association between low serum level of anti-inflammatory cytokines and the TB and MDD comorbidity.” Please, clarify and modify the conclusion.

Author Response

We appreciate sincerely for the time devoted to our work. The observations allowed to improve considerably the manuscript. Thank you.

Comments: 

Abstract: it is suitable. However, I suggest that conclusion be rewritten because I understood that the last paragraph from Introduction described better those findings of the present study. In addition, it would be interesting to comment briefly in the Abstract the importance of the study in the research field. 

Answer. We appreciated the comment, so we revised the conclusion of the Abstract and eliminated the paragraph that was. Page 1, lines 33-34. In the new version, we write the following paragraph: A high frequency of IFN-gamma-producing cells was associated to low levels of anti-inflammatory cytokines in MDD-TB patients. P1, lines 34-36. In response to your suggestion we inserted the following paragraph: Therefore, an integrated clinical practice should be considered. P1L21.   

Introduction and Discussion: good and objetive.

Materials and Methods: 1. How the number of patients and healthy controls were defined for the casuistic?

Answer.  The work design characteristics are of a descriptive, preliminary, and transversal one. The number of patients and controls were defined according to requirements of statistical analysis.

2. Were PBMC stained with live dead to exclude viable from non-viable cells assessed by flow cytometry?

Answer. In fact, we do not mention the use of 7-aminoactinomycin-D (7-AAD) in flow cytometry analysis. We only used 7-AAD to standardize the immunofluorescence method. After cell culture, cells were harvested, washed, and stained by fluorescent antibodies to CD3, CD4, CD8, CD69, and CD25 for the purpose of confirming both activation and percentages of T cell subpopulations. For these assays, we added 7-AAD prior to flow cytometry to discriminate non-viable cells from viable cells. However, in subsequent assays, we no longer use 7-AAD because cells were permeabilized by saponin to assess the frequency of intracellular cytokine-positive cells. In the new version, we eliminated the acquisition of 7-AAD. P3L97.

Results:

3.  The authors  could show the gate strategy used for flow cytometry, and a representative analysis obtained from the more significant results. 

Answer. We appreciate your comment, but unfortunately, the flow cytometry files remain in possession of the Research Section of national Institute of Respiratory Diseases where the study was managed. The access to those files would take more time.

4. Why did the authors use CD69 and CD25 staining?. They did not show that results and did not mention about them. Please, explain

Answer. We used fluorescein isothiocyanate (FITC)- or phycoerythrin (PE)-labelled antibodies to CD69 and CD25, because theses moleculaes are considered membrane markers of lymphocyte activation. However, we observed in PBMC that the percentage of CD25-positive cells was lower than those CD69-positive cells after activated by PMA plus ionomycin. In order to increase the expression of CD25, the use of plate-immobilized anti-CD3 and soluble CD28 antibodies is required. Hence, we decided to make a gate for CD69-positive cells, since this marker remains expressed for up to 24 hours. From the gate of CD69-positive cells, the analysis of cell positivity to TNF-alpha, IFN-gamma, IL-10, and IL-12 was performed. In the new version, we inseted a paragraph that briefly explains the use of antibodies to CD69 or CD25. Likewise, we explained that positive cells for intracellular cytokines were counted in a CD69-positive cell gate. P2L96 and P3L 131-134.

5. Bar graphs should be representative also by individual data plus bar.

Answer. We attach the from figures 1, 3, 4, 7, and 8, where the individual data are represented. if the Reviewer consider that the individual date graphs should be added to bar graphs, we have not problem.

6. Although PBMC from MDD-TB exhibited higher IFN-gamma-producing cells compared to MDD or TB samples, those frequencies were not different from PBMC from healthy donors. How do the authors explain that?. In the Discussion, the authors described: "The increased frecuency of IFN-gamma-producing cells from MDD-TB patients might be indirectly due to deficient regulation caused by low levels of anti-inflammatory cytokines, such as IL-10, IL-4, and IL-13. An excess of inflammatory can contribute to increased tissue damage during the immune response against mycobacterial infection [34]." Might the authors describe "an excess of inflammation" when the cytokine levels in the serum fom MDD-TB patients were lower than those in MDD and similar to TB patients?

Answer. The condition of the MDD-TB patients is complex, due to association of two nosological entities. This appears to increase the proportion of IFN-producing cells capable of responding to the infectious agent. Thus, it could be considered a one of the findings of this work. Regarding to comment on the Discussion. After reanalysing our data, we found that is not possible to state that there is an excess of inflammation due to the lack of supporting evidence. Therefore, we decided to eliminate the concept "an excess of inflammation" in the manuscript. P11,L 319-320, and P12, L 421.

7.  How do the authors explain high cytokine-producing cells and low level of serum cytokines in the MDD-TB group?

Answer. The MDD-TB group exhibit only a high proportion of IFN-gamma-producing cells, since the other cytokine-producing cells do not show statistical differences among the groups, except for MDD group. With respect to low levels of serum cytokines in the MDD-TB group, one possible explanation is because they are being used. Cytokines are consumed rapidly especially in an infection. Nevertheless, there are no statistical differences among the serum cytokine levels from the MDD-TB, TB and control groups, but not for IL-13. Only the amount of serum cytokines from the MDD group shows differences as compared with other study groups. In contrast, the chemokines, such as MCP-1 and IL-8 from MDD-TB and TB groups exhibited lower serum levels than the control group, perhaps for the same reason explained above.

8. The results are difficult to interpret because assays were performed with polyclonal stimulation and not with mycobacteria-specific antigen. This point should be discussed. Authors need to discuss the limitations of the study.

Answer. True, the results are dificult to interpret, and further assays are needed to clarify the immunological status of MDD-TB patients. However, it is a preliminary study, which has its limitations, such as the ones you point out. We agree with you in adding the limitations of our study. In the new bersion, we insert a paragraph where the limitations of the study are mentioned. P11, L 323-326.

Conclusion: too extensive

Answer. We have shortened Conclusions by removing paragraphs that contained redundant and speculative information. P11, L 333-339.

9. The authors could address straight the take home message.

Answer. We consider making a brief conclusion and then directly address the take home message.

10. The findings do not show that: "However, the exception was a high frequency of IFN-gamma-producing stimulated cells from MDD-TB patients, which was consistent with a powerful association between low serum level of anti-inflammatory cytokines and the TB and MDD-TB comorbidity." Please, clarify and modify the conclusion.

Answer. True, we have not evidence. Thus, we decided to eliminate the paragraph. P11, L 337-339.

Reviewer 2 Report

This is an interesting article and appears to be well executed. My only concern is that the conclusions are not very exciting and the manuscript is mostly descriptive -  it would be interesting to know for example what were the results of the patients before being infected with TB but that would require a different study - like a cohort - something I understand is not possible for the authors. Overall this is a scientifically serious work and deserves to be published as it supports the mounting evidence in the field - it can help and guide others in the future.

Author Response

Dear Reviewer,

We appreciate sincerely your comments about our work. Certainly, we agree with you that our study is descriptive type, which does not allow to stablish a cause-effect relationship. However, your observation is very interesting since could improve this research field. In addition, in the checklist, you market that the conclusions could be improved. For this reason, we substitute the paragraph of conclusions in the Abstract, hoping that new paragraph will be clearer. Now, the conclusion is "A high frequency of IFN-γ-producing cells was associated to low levels of serum anti-inflammatory cytokines in MDD-TB patients". 

Kind regards,

Ricardo Lascurain, Departamento de Bioquímica, Facultad de Medicina, Universidad Nacional Autónoma de México.